Convolutionary, Evolutionary, Revolutionary: What's next for Bodies, Brains and AI?

Peter Stratton[1,2]
1. The University of Queensland, St Lucia, QLD, Australia
2. Massachusetts Institute of Technology, Cambridge, MA, USA

Abstract: In recent years we have made significant progress identifying computational principles that underlie neural function. While certainly not complete, we have sufficient evidence that a synthesis of these ideas could result in an understanding of how neural computation emerges from a combination of innate dynamics and plasticity, and which could potentially be used to construct new AI technologies with unique capabilities. I discuss the relevant principles, the advantages they have for computation, and how they can benefit AI. Limitations of current AI are generally recognized, but fewer people are aware that we understand enough about the brain to immediately offer novel AI formulations.

## Principles of neural function and plasticity.

The flexibility, adaptability and resilience of even simple brains are unmatched by any current technology. The recent unexpected difficulties in realizing truly autonomous vehicles, making reliable medical diagnoses, detecting offensive online content, creating useful chat-bots and even just recognizing faces, show that brains remain functionally more capable than we can currently emulate. Significant differences exist between spiking neural networks (SNNs) and the superficially-similar artificial neural networks (ANNs) used for Deep Learning (DL), and in these differences can be found potential clues to the next AI revolution:

- Real neurons communicate with spikes where spike timing forms an integral part of the neural code[1]. The computational benefits of sparse spike coding are substantial[2]. SNNs are also rigorously more powerful than their real-valued counterparts[3].
- Networks self-organize to represent feedforward input structure[4]. The mechanisms the brain uses to accomplish this have been established (spike timing dependent plasticity (STDP)[5-7], several homeostatic mechanisms[5,6] and local decorrelating inhibition[5-7] together implement sparse non-negative matrix factorization[6,8]).
- Abundant feedback connections build predictive models[9-11] by learning to invert the self-organized feedforward representations (also using STDP[11]).
- Oscillations[12,13] and quasi-chaotic transitions[14,15] continuously reconfigure neural circuits based on the needs of the task at hand. Activity levels, oscillation frequencies and proximity to phase boundaries are controlled through recurrent thalamic projections[16].
- Multi-scale inhibitory mechanisms[7,13,16,17] ensure only best-matching circuits are engaged for any given task, implementing powerful *k-winner-take-all* computations[18].
- Spike conduction delays[13,19], oscillations[13] and short-term plasticity (STP)[19,20] innately represent time in the brain[19,20]. In recurrent neural circuits, STP can maintain memory states (working memory) for indefinite lengths of time[21].
- Dopamine modulates the gain of STDP for model-free reinforcement learning (RL)[22]. Other neuromodulators have arguably equally important effects – acetylcholine increases the efficacy of feedforward connections and attention to inputs, noradrenaline responds to novel and salient inputs, and serotonin to risks and threats[23].

The above principles are well-established, even if all are not yet universally accepted. Other principles are more speculative, or the specific underlying mechanisms are not fully established:

- Brains combine predictive models of the world with oscillations and dynamic circuit reconfiguration to create internalized simulations of 'what if' scenarios and future plans[24,25]. These models also flag unexpected and out-of-distribution events[26].
- Closely related to the above is the idea of stochastic sampling[11,27]. This same process probably applies to outputs (actions), including sampling from possible *sequences* of actions (equivalent to 'what if' simulations).
- Dopamine works with oscillations, dynamic circuit reconfiguration and the internal world models to implement model-based RL. How this occurs is unclear, but prediction of reward and temporal-difference (TD)-style learning will be essential components[28].
- Explicit actions are just the final steps in a series of neural events learned through reinforcement – i.e. actions are preceded by sequences of internal neural (cognitive) operations that, from the perspective of neural activity patterns and TD learning, are indistinguishable from those patterns that directly cause movement of the body in the world. High level cognitive functions are therefore simply 'internal actions'[25].
- Subcortical circuits, particularly the basal ganglia, cerebellum, brainstem and spinal cord, fully control well-trained movements and serialization of all other motor outputs. These regions may use partially different mechanisms but are still tightly integrated[29].
- Computational principles apply similarly across all of cortex. Differentiation of function occurs predominantly through structural connectivity.

To compute means to control the flow of information, and to store, recall, organize, integrate and transform information in pursuit of a defined outcome or ongoing effect. In the case of the brain, it also means to flexibly adapt to unforeseen conditions in ways that no computers can currently achieve. The brain accomplishes this by flexibly activating neural assemblies (groups of neurons) in combinatorial patterns that best represent the confluence of sensory input and current internal state. But what controls which assemblies should be active? Neural assemblies respond when they recognise (i.e. are keyed by) particular afferent spike patterns, and only when this key-match is good enough to win the multi-scale-inhibition competition. The vital insight is that neurons and assemblies respond *when they are required* and *without centralized control*. This fortuitous outcome is the result of the brain's multiple plasticity mechanisms that bias the dynamics towards activity that is ultimately rewarded[30], while extracting features that predict reward, and maintaining activity within useful dynamical bounds.

Neural assemblies, dynamics, cognition and creativity.

When an assembly becomes active in response to a recognized input, it outputs a transformation of its inputs that is shaped by self-organization and reward learning. In so doing it is performing a computation on its inputs that meets either innate (self-organizing) or external (reward-bearing) criteria. Neural connections are convergent, divergent, hierarchical and re-entrant, supporting spatiotemporal activity patterns that are exquisitely intricate and inter-dependent. Assemblies couple in novel patterns in response to novel inputs, and exploit the chaotic nature of the transitions between states to form novel patterns any time, in a manner related to binding of representations, fluid intelligence and creativity[31]. However they are far from random; constrained by neural architecture and shaped by plasticity, they are finely honed to be task-specific. These patterns underlie the combinatorial computational power of the brain.

The brain does not follow a program. Brain regions do not encode packets of information which are transmitted to receiving regions for decoding and processing, and brains do not work 'despite the noise'. Due to efficient coding and stochastic sampling, what we are tempted to think

of as noise is in fact the entire computation[11]. Engineering-style reductionist simplifications yield no insights into neural function. The brain is the ultimate bootstrapped physical dynamical system. A neuron simply sits and listens[32]. When it hears an incoming pattern of spikes that matches a pattern it knows, it responds with a spike of its own. That's it! Repeat this process recursively tens to trillions of times, and suddenly you have a brain controlling a body in the world or doing something else equally clever. Our challenge is to understand how this occurs. **We require a new class of theories that dispose of the simplistic stimulus-driven encode/ transmit/decode doctrine. We must embrace the brain's inherent dynamic complexity and emergent properties, and explain how plasticity molds the dynamics to capture useful couplings across brain regions and between the brain, the body and the world. My contention is that the above principles are sufficient to meet this challenge at some level, in order to both better understand the brain and to construct better brain-inspired AI.**

Brains to AI.

The differences between brains and ANNs lead to significant concrete differences in capabilities:

- AI is difficult to train and typically requires huge amounts of clean labelled data. Due to its ability to discover parts-based representations and combine them in novel patterns, the brain implements transfer learning by chunking often with just a handful of training samples.
- AI systems have enormous energy requirements for training and operation. Due to sparse efficient spike-time coding, the brain runs on less than the power of a light bulb.
- AI systems need to be pre-trained and any new information typically requires complete re-training. The brain learns online continuously using transfer learning and dynamical processes (activity patterns) for integrating that knowledge into long-term networks.
- AI is terribly brittle and can be easily fooled by adversarial input that needs to be shifted only slightly outside the training distribution. Brains generalize exceptionally well due to modular self-organization, predictive feedback and transient combinatorial dynamics.
- AI systems can perform only the task for which they are trained. Due to its ability to dynamically reconfigure through oscillations and internal actions, the brain can perform multiple tasks and switch between them as required.

Brains generate explanatory causal models using STDP, predictive feedback and working memory. What we currently call AI is fundamentally still big data and correlation analysis, predominantly used to generate classifications and occasionally predictions. There are exceptions to this rule[33,34]; interestingly these exceptions tend to draw inspiration directly from the brain in order to improve on the capabilities of DL. While improvements are often achieved, the insights are applied in piecemeal fashion and many of the compelling advantages of neural processing remain ignored and unharnessed. While recurrent ANNs are Turing complete, we know from experience with DL that choice of architecture and how information is represented make a difference, and that just because a task can theoretically be performed does not mean that it can be done efficiently, or even that it can be learned at all. SNNs are more powerful than ANNs of equal size, and are dynamically and architecturally ideal for representing spatio-temporal patterns and for building causal models of the world. It is reasonable to expect that there will be classes of problem, relevant for our usage of AI, for which ANNs will fail in practice but that can be learned and performed efficiently by SNNs.

Studies have already shown how deep networks implemented with spiking neurons outperform standard DL in some respects. On simple problems they require orders of magnitude fewer

training samples, they use unlabelled training data, and they generate sparse efficient parts-based representations[35]. These early results are significant but they reveal only a small subset of the full capabilities of brains and SNNs. DL requires large quantities of training data because typically the full range of input space needs to be explicitly covered during training. Energy requirements for training state-of-the-art deep networks are already measured in megawatt-hours and the curse of dimensionality is causing an exponential increase as ever-larger problems are tackled; clearly this is an unsustainable trajectory. While parts-based decompositions and generative models can also be implemented using DL, these functions are parsimoniously implemented in SNNs by STDP. The fundamental nature of spikes as momentary events additionally leads to powerful temporal representations, rapid processing, and intrinsic dynamics that allow for stochastic sampling and dynamic reconfiguration of neural circuits to match ongoing computational needs. DL offers few of these capabilities. Complex neural dynamics mirror the complexity evident in the real world – brains are complex precisely because the world is complex[36] ('complex' is used in the complex systems sense and does not just mean 'complicated'). Indeed neural dynamics continuously and task-dependently couple both with the body and with sensory events[37]. The clear implication here is that brains and SNNs are ideally suited for embodied interactions with the real world, and the full advantages of SNNs will become evident when these neural computational principles are applied to real-world problems. Notably, these are exactly the kinds of problems for which DL is having trouble scaling.

The true computational power of the brain lies in the synergistic integration of all the principles of neural computation[11,30]. To the author's knowledge, such an integration has never been attempted at scale. Oscillations and spike-time coding have rarely, if ever, been combined with RL to flexibly route information through a neural network, for example. Further combining such a network with self-organizing plasticity could then create a network that can generalize and respond flexibly to new inputs; feedback could allow for attention to unexpected inputs; and so on. There is a feeling in the ANN community that the dynamics of spiking networks are difficult to conceptualize and control. On the contrary, it is quite possible that the synergistic combination of principles will lead to intuitive dynamics and a deeper understanding of the underlying mechanisms. **All the principles that have been discussed here are mechanistic, evidence-based, and *realistically implementable in neural circuit models and AI prototypes.* It is probable that revolutionary computational systems can be created in this way with only moderate expenditure of resources and effort.**

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

Engineered systems, including our most advanced Artificial Intelligences (AIs) driven by Deep Learning (DL), are typically characterised by 1) their suitability for only the specific purposes for which they were designed, 2) their inability to operate autonomously in unconstrained real-world environments, and 3) the increasing engineering effort required as systems become ever more complex and purpose-built.

The brain's flexibility, and indeed its entire computational capacity, is rooted in the activity dynamics of its components[1,2] – that is, in neurons connected by synapses into networks. DL pioneers LeCun and Hinton have proposed that supervised learning will ultimately need to be abandoned, and that unsupervised learning, as occurs in the brain, is the way forward to more capable AI[3,4]. An understanding is emerging that[5]: "The brain seamlessly merges bottom-up discriminative and top-down generative computations in perceptual inference, and model-free and model-based control... [We must] explain task performance on the basis of neuronal dynamics and provide a mechanistic account of how the brain gives rise to the mind." Buzsaki states more succinctly[2]: "Brains do not process information: they create it."

This hypothesis has significant implications for results of experiments that purport to find low-dimensional manifolds in population firing rates, for example in diffusion models of decision making. Working memory combined with stochastic sampling only gives the appearance of a simple low-D representation. The brain is not accumulating evidence in the firing rates of a large neural population; this would be exceedingly inefficient (a low-D manifold on low-D firing rates). Instead, it is doing stochastic sampling of the possible interpretations of the input. The afferent evidence from which it is sampling is also not held in firing rates; it's held in the states of network activity in lower regions (which themselves are doing stochastic sampling of evidence from regions below them). As the evidence is accumulated (probably in STP, not firing rates), firing rates increase because the probability distributions are narrowing and the sampling is being constrained. The network activity that we interpret as firing rates is actually a series of complex information-rich spatiotemporal patterns that store a huge amount of information about the context, stimulus, upcoming response and ongoing self-generated brain state.

According to Friston[1]: "By studying the dynamics and self-organization of functional networks, we may gain insight into the true nature of the brain as the embodiment of the mind." Structurally, the brain contains large overlaps between its functional modules and strong cross-connections between hierarchies at all levels. Such anatomical structure causes complex patterns of competition and coupling that interact through and across multiple hierarchical levels simultaneously. Active assembly boundaries are fluid, ranging in size from a handful of neurons up to large regions, and no single module is ever at the top, or in control, from the perspective of either static connectivity or dynamic activity. Neural computation therefore manifests as a continuous superposition of transient dynamic states, flexibly mediated in the short term by opposing forces of competition and coupling within and between neural assemblies and the world, and in the long term by self-organization and reward.

Assemblies are activated by input that they recognize. Output spikes that are generated by an active assembly add to the input spike patterns being received by other neurons, and influence their activities to modulate and further direct the computations that are performed. Once the particular input that activated an assembly disappears, or the brain state changes, input to the assembly no longer matches, and the assembly naturally shuts down until it next receives keying input. This mechanism has the effect of always finding a part of the brain to process any given

input or brain state – when the key-match is good the brain responds quickly, driving lateral inhibition and pre-emptively shutting down other neurons and regions which might otherwise have responded. If the match is poor the brain responds more slowly since a poor match needs longer to drive neurons to threshold. STDP, RL and homeostatic mechanisms cause the recruitment of more neurons to represent common inputs and well-trained tasks through the following mechanism: Commonly-occurring inputs will cause excessive firing of the associated assemblies which will then homeostatically raise their thresholds to reduce their firing rates. This will give other neurons that were previously inhibited by lateral inhibition from those assemblies a chance to respond instead, and when they do STDP will then solidify their new roles in representing the input. This recruitment process increases the fidelity and discriminability of representations of common inputs and also increases processing speed due to finer and better key-matches.

I am not advocating for a biophysically-detailed bottom-up approach nor a top-down cognitive model. This is 'sideways-in', where relevant biophysical principles are abstracted and combined in such a way as to bring about emergence of function as occurs in the brain. The primary modelling level (the level that is modelled explicitly) is the level of neurons, synapses and spikes. At the level below are ion channels, neurotransmitter release, synaptic currents and membrane dynamics – these are abstracted and modelled as mathematical functions rather than explicitly. At the level above are populations of neurons, oscillations and network dynamics – these emerge from interactions of components at lower levels. The central modelled level is therefore one level below the emergent properties of interest. Spikes and channel dynamics are non-complex even when they are stochastic. The complex functional properties of the brain emerge at the level of networks and population dynamics.

Each of the computational principles – spike-time coding, self-organization, short term plasticity, reward learning, homeostasis, feedback predictive circuits, conduction delays, oscillations, innate dynamics, stochastic sampling, multi-scale inhibition, winner-take-all, and embodiment – are research topics that have been separately investigated, some quite extensively. However, a rich understanding of neural function can only be obtained by understanding how these principles synergistically combine[6-8]. Next-generation AI using these principles will inherit the many advantages of directly brain-inspired neural processing. If similar attention is given to these SNN mechanisms as has been given to ANNs over the last 10 years, it seems reasonable to expect that next-gen AI can be realized, or at the very least extensive progress in this direction can be made.

## Appendix References.

1       Park, H.-J. & Friston, K. Structural and functional brain networks: from connections to cognition. Science 342, 1238411 (2013).
2       Buzsaki, G. The Brain from Inside Out.  (Oxford University Press, 2019).
3       LeCun, Y., Bengio, Y. & Hinton, G. Deep learning. Nature 521, 436 (2015).
4       Hao, K. https://www.technologyreview.com/s/613954/the-next-ai-revolution-will-come-from-machine-learnings-most-underrated-form/, 2019).
5       Kriegeskorte, N. & Douglas, P. K. Cognitive computational neuroscience. Nature Neuroscience 21, 1148-1160 (2018).
6       Savin, C. & Triesch, J. Emergence of task-dependent representations in working memory circuits. Frontiers in Computational Neuroscience 8 (2014).
7       Hartmann, C., Lazar, A., Nessler, B. & Triesch, J. Where's the noise? Key features of spontaneous activity and neural variability arise through learning in a deterministic network. PLoS Computational Biology 11, e1004640 (2015).
8       Lazar, A., Pipa, G. & Triesch, J. SORN: a self-organizing recurrent neural network. Frontiers in Computational Neuroscience 3, doi:10.3389/neuro.10.023.2009 (2009).
