# OpenReview forum: "Convolutionary, Evolutionary, Revolutionary: What’s next for Bodies, Brains and AI?"
_NeurIPS.cc/2019/Workshop/Neuro_AI — Real Neurons & Hidden Units @ NeurIPS 2019 Poster_

### Official Review · AnonReviewer1 · 2019-09-20
**A fun read and good summary of the differences in our approaches to undersanding and building natural and artificial intelligence**

**Clarity:** 4

**Comment:**

A useful discussion piece.

I think the main value of the piece for this workshop is in using the points made about spiking and dynamic neural nets as sources for models in AI. The piece could be more explicitly structured around this. I.e. start with the list of shortcomings in AI (currently in the last section), and then follow up with how features of biological neural networks and brains could solve these problems. Currently it's structured the other way around.

**Category:**

Neuro->AI

**Clarity Comment:**

This is clearly written. I found no large points to improve.

**Evaluation:**

4: Very good

**Importance:**

4: Very important

**Importance Comment:**

Thinking conceptually about the shortcomings of both current AI approaches, and models of brains is important work. What is often framed as a data analysis problem in neuroscience ("we now have all this great data, what do we do with it?") is also a more conceptual problem. I agree with the author's view that moving beyond a static encoding/transmit/decode picture is helpful for both neuroscience and models of AI. Prompting discussion in this area is useful.

**Intersection:**

4: High

**Intersection Comment:**

This is relevant to both communities to think about.

**Rigor Comment:**

I didn't find many large omissions in important details or misrepresentations of the literature. Two points would be:

Can the list of shortcomings in AI be more explicitly linked to the list of properties of SNNs and how these would address them? There is a response that ‘the brain doesn’t have this problem’... but it doesn’t go on to say exactly how the list of SNN properties could help solve it in the AI case.

Responding to the comment that "The true computational power of the brain lies in the synergistic integration of all the principles of neural computation. To the author’s knowledge, such an integration has never been attempted at scale."
It seems reasonable to not count the large-scale brain simulation projects (e.g. blue brain) here (as they do not include any behavior!). But I think Eliasmith’s work on Spaun is an attempt to do something like what you're describing at a large scale. It doesn’t include learning, but is otherwise a holistic attempt relevant for both neuro and AI.



**Technical Rigor:**

4: Very convincing

---

### Official Review · AnonReviewer3 · 2019-09-21
**Interesting but disjointed**

**Clarity:** 2

**Comment:**

An interesting perspective, but one that would have much broader impact and appeal if the argumentation were tightened and more concrete examples of next-steps in AI were provided.

**Category:**

Neuro->AI

**Clarity Comment:**

As outlined above, I find the style of argumentation somewhat opaque for those outside the immediate area of study. More didactic explanation of how observed phenomena being cited link to the broad views of neural function would go a long way towards strengthening the presentation.

**Evaluation:**

3: Good

**Importance:**

3: Important

**Importance Comment:**

The author(s) raise important known biological phenomena, largely centered around temporal dynamics, that are not well-incorporated into existing AI approaches. They argue that these dynamics will be critical for advancing AIs.
This point is an important and valuable one. However the overall importance of the piece is somewhat reduced by the disjointed presentation and lack of explicit links between broad ideas into concrete examples.

**Intersection:**

4: High

**Intersection Comment:**

A key strength of the manuscript is its treatment of questions relevant to the interface of neuroscience and AI. It is very topical for this workshop.

**Rigor Comment:**

The paper provides a very wide overview of neural computation, and does not provide a formal or didactic mechanism to link all ideas and concepts together. The list of observed biological phenomena summarized in "Principles of neural function and plasticity" and following paragraph are not well-linked to form an argument with concrete evidence to my reading. While I do not necessarily disagree with the broad statements made, I find the style of argumentation lacking in sufficiently rigorous treatment, particularly for the sections on biological computations. Similarly, the sections on AI lack citations to support statements. While I understand they are commonly-observed phenomena, a more grounded treatment would improve the manuscript. Providing concrete examples of problems that would benefit from models incorporating dynamics, for instance, would greatly improve the impact and broaden the audience.

**Technical Rigor:**

2: Marginally convincing

---

### Official Review · AnonReviewer2 · 2019-09-26
**An opinion on the promise of current thinking in comp-neuro to revolutionize AI: overstated, but food for thought**

**Clarity:** 3

**Comment:**

While it covers important ground, I think the arguments need more refinement and focus before they can inspire productive discussion.

**Category:**

Neuro->AI

**Clarity Comment:**

While it covers important ground, I think the arguments need more refinement and focus before they can inspire productive discussion.

Its more a series of statements than a cleverly woven argument.  But the individual statements are sometimes seductive.

For example ...

"A neuron simply sits and listens. When it hears an incoming pattern of spikes that
matches a pattern it knows, it responds with a spike of its own. That’s it! Repeat this process
recursively tens to trillions of times, and suddenly you have a brain controlling a body in the
world or doing something else equally clever. Our challenge is to understand how this occurs.
We require a new class of theories that dispose of the simplistic stimulus-driven encode/
transmit/decode doctrine. "

The devil is in the details, the "how" of "suddenly".

I feel this statement:
"Our challenge is to understand how this occurs.
We require a new class of theories that dispose of the simplistic stimulus-driven encode/
transmit/decode doctrine. "

Largely contradicts this one

"It is probable that revolutionary computational systems can be created in this way with only
moderate expenditure of resources and effort"



**Evaluation:**

2: Poor

**Importance:**

2: Marginally important

**Importance Comment:**

 The paper provides a broadly useful synthesis of key differences between ANN and SNN approaches.  However, the multiple grandiose statements, and some that are downright misleading left me puzzling what I learned.

Its an opinion piece.  It offers a call to action to do more comp-neuro, in that it could revolutionise AI.




**Intersection:**

4: High

**Intersection Comment:**

I felt the paper could have done more to link with current state-of-the-art AI approaches.  There was an absence of nuance.

**Rigor Comment:**

The paper opens

"In recent years we have made significant progress identifying computational principles
that underlie neural function. While not yet complete, we have sufficient evidence that a
synthesis of these ideas could result in an understanding of how neural computation emerges
from a combination of innate dynamics and plasticity"

What follows is a useful survey of a selection of ideas, by far not complete.   For example, many of the interactions between myriad excitatory and inhibitory types across brains regions and neuromodulators, of which dopamine is just one of several, is largely unknown.  Arguably ACh and noradrenaline are more important for network states and dynamics, and equally important for plasticity as dopamine.  The dynamics of neuromodulation is largely unknown.

Which leads me to a few concerns.

"It is probable that revolutionary computational systems can be created in this way with only
moderate expenditure of resources and effort"

Of course whole fields are working on this problem.  Hardly what I'd call moderate effort.

Claims of efficiency of more brain-like approaches compared to AI are disingenuous. A major draw-back of spiking models is that they are much more costly than ANNs, because of the small time-steps required. Sure neuromorphic systems are coming, but not definitely not with moderate expenditure of resources and effort"







**Technical Rigor:**

2: Marginally convincing

---

### Decision · Program_Chairs · 2019-10-02

Accept (Poster)